# Influence of Crack Geometry on Dynamic Damage of Cracked Rock: Crack Number and Filling Material

**Feili Wang** [†], **Shuhong Wang** * and **Zhanguo Xiu** *,[†]

School of Resource & Civil Engineering, Northeastern University, Shenyang 110819, China; wangfeili109@126.com
* Correspondence: wangshuhong@mail.neu.edu.cn (S.W.); xiuzhanguo109@126.com (Z.X.)
† The authors contributed equally to this work.

**Abstract:** The dynamic damage of cracked rock threatens the stability of rock structures in rock engineering applications such as underground excavation, mineral exploration and rock slopes. In this study, the dynamic damage of cracked rock with different spatial geometry was investigated in an experimental method. Approximately 54 sandstone specimens with different numbers of joints and different filling materials were tested using the split Hopkinson pressure bar (SHPB) apparatus. The energy absorption in this process was analyzed, and the damage variable was obtained. The experimental results revealed that the dynamic damage of cracked rock is obviously influenced by the number of cracks; the larger the number, the higher the energy absorption and the bigger the dynamic damage variable. Moreover, it was observed from the dynamic compressive experiments that the energy absorption and the dynamic variable decreased with the strength and cohesion of the filling material, indicating that the filling material of crack has considerable influence on the dynamic damage of cracked rock.

**Keywords:** cracked rock; geometric characteristics; split Hopkinson pressure bar (SHPB); energy absorption; damage variable

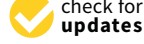



## 1. Introduction

With limited land and mineral resources, mining and geotechnical engineering are continuously being developed; however, this development has brought increasing challenges. A large number of geological disasters have been induced by dynamic disturbance [1,2]. Many studies have indicated that the dynamic damage of cracked rock bears significant adverse impacts and can lead to geological disasters. The strength deterioration and deformation of cracked rock are the main triggers of geological disasters [3–5]. Understanding the dynamic properties of cracked rock is thus necessary because it helps to prevent the potential for geotechnical failures.

Cracks generally exist in natural rock masses, and their physical and geometrical characteristics have considerable influence on the properties of cracked rock, such as surface roughness, filling material, rock type and crack length [6–13]. It is well accepted that the dynamic properties of cracks control the dynamic damage of cracked rock [14]. Moreover, the propagation of stress waves in cracks accelerates the failure of cracked rock. Many publications focus on the influence of filling materials on wave attenuation [15–20]. The research results show that the attenuation of a stress wave is strongly affected by its reflection and transmission at the interfaces as well as by the dynamic properties of filling materials. The number of cracks is also an important geometrical characteristic influencing wave propagation [21–24]. From previous studies, we can see that the transmission coefficient across fractures is a function of the fracture spacing for different numbers of cracks. The presence of cracks in a rock mass is clearly of paramount importance as it dramatically affects the dynamic behavior of cracked rock.

John Hopkinson and his son, Bertram Hopkinson, invented a pressure bar to obtain the pressure-time curve with the dynamic load exerted by detonation (Hopkinson, 1914). The Hopkinson bar was further developed by Kolsky (1949) into the split Hopkinson pressure bar (SHPB). SHPB has become a reliable high strain rate loading apparatus to measure the dynamic response of brittle materials under impact loadings. It has been suggested recently as a standard method to measure the dynamic mechanical properties of rocks by the International Society for Rock Mechanics and Rock Engineering (ISRM). As a widely used device to quantify the dynamic properties of various brittle materials at high loading or strain rates, the SHPB system is used to study the compressive response [25,26], tensile failure [27–29], shear strength [30–32] and fracture characteristics of rocks [33,34]. There is no doubt that the SHPB system is a technique available to measure the dynamic compressive response of cracked rock under impact loadings.

The previous studies on creaked rock focused primarily on the significant effects of cracks' physical characteristics on the dynamic properties. Research on the quantitative relation between the dynamic damage of cracked rock and the spatial geometry of cracks is limited. Therefore, this study focused on the dynamic damage of cracked rock with different spatial geometry. The influences of the number of joints and filling material on the dynamic properties of cracked rock were tested by the split Hopkinson pressure bar (SHPB) apparatus. The energy absorption in the dynamic impact process was analyzed, and the damage variable was obtained.

## 2. SHPB Test

### 2.1. Experimental Setup

An SHPB system was adopted to systematically investigate the dynamic behavior of cracked rock. The configuration of this apparatus is schematically shown in Figure 1. The dynamic loading system, containing a high-pressure gas source, a gas gun, a striker, an incident bar and a transmitted bar, could dynamically load the specimen. All bars were made of a superior alloy steel material with a Young's modulus of 206 GPa and a longitudinal wave velocity of 5122.698 m/s. The striker bar had a length and diameter of 600 mm and 100 mm, respectively, while the length and diameter of the incident bar were 5000 mm and 100 mm, respectively.

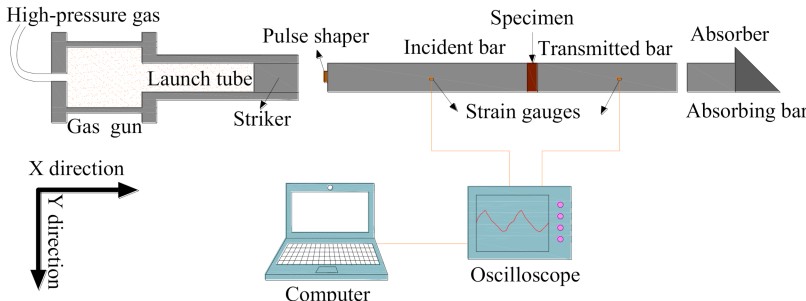

**Figure 1.** Schematics of a split Hopkinson pressure bar (SHPB) apparatus.

During the tests, the strain pulse signal was captured using a Wheatstone bridge and a pair of strain gauges (Type: BE120-5AA, Resistance: 120.4 ± 0.1, Gage factor: 2.20 ± 1%) symmetrically mounted on the incident and transmitted bar surfaces across the bar diameter. The schematic showing the position of the strain gauges is given in Figure 2. It indicates that the distance between the strain gauge and the specimen on the incident bar was equal to that on the transmitted bar, i.e., 2500 mm. Thereafter, the signal was amplified and then recorded by an amplifier and an oscilloscope. In this study, an eight-channel digital oscilloscope was applied, and the sampling rate was set at 130 kHz.

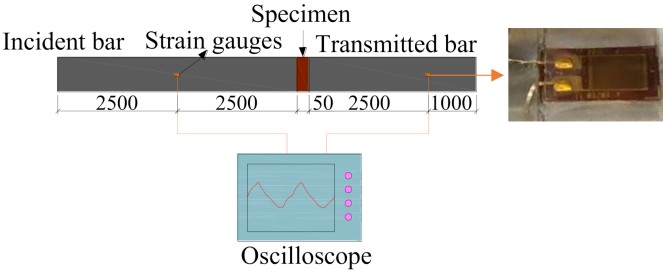

**Figure 2.** Strain gauges used in SHPB testing system.

Generally, careful pulse shaping is crucial to dynamic tests conducted on the SHPB system for brittle materials. Applying this technique to the SHPB system can generate a ramp incident pulse rather than a trapezoidal one, facilitating the force equilibrium in a dynamic test [35,36]. Among all the methods, the pulse shaper is a relatively simple one for pulse shaping. A pulse shaper made of AC1100 copper and 50 mm in diameter and 1.0 mm in thickness was thus employed in this study. Before a single test, a shaper was centrally attached to the impact end of the incident bar.

For a single dynamic test, specimens were assembled between the incident bar and the transmitted bar. To minimize the interfacial friction on both ends of the specimen, Vaseline was used as a lubricant. Before the specimen was sandwiched between the incident and transmitted bar, Vaseline was applied on both ends of the specimen. When the specimens were assembled, the striker was launched by the sudden release of high-pressure gas. After the striker impacted the incident bar, an incident pulse was generated and propagated through the bar. Owing to the mismatch of wave impedance between the bar and the specimen, part of the incident pulse was reflected at the rock-steel interface. At the same time, the rest of the incident pulse was transmitted into the specimen and the transmitted bar [37]. The specimen was thus dynamically loaded.

### 2.2. Specimen Preparation

Red sandstone was chosen as the testing material for the dynamic test. Three main steps were used to prepare the cracked rock. First, cylindrical specimens with dimensions of 100 mm × 50 mm (diameter × length) were carefully prepared. For consistency, all cylindrical specimens were made of the same sandstone block. Second, the size and location of cracks were digitalized and reconfigured using computer-aided design (CAD) techniques. In this study, the rectangular crack parallel to the longitudinal axis of the cylindrical specimen was designed with the dimensions of 50 × 40 × 4 mm. Subsequently, the cylindrical specimens were cut and engraved following the schematics of the specimen and using a computer numerical control (CNC) water-jet machine to obtain the cracked rock specimen. In this study, the man-made crack in the specimen was filled. The process of filling the crack was as follows: (1) Five types of filling materials were prepared: soil, sand, and materials with the weight ratio of soil/sand = 2, 1, and 0.5. (2) The filling material was poured into the crack three times. After each pour, the specimen was placed on the vibration table for 3 min of vibration. The schematics of the specimen with one crack is shown in Figure 3.

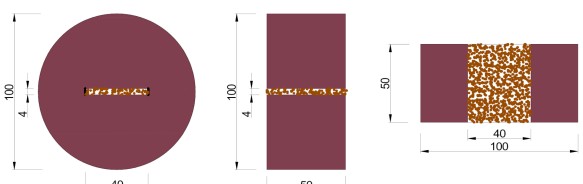

**Figure 3.** Schematics of the specimen with one crack.

To further study the properties of the filling materials (i.e., ISO sand and Liaoning soil), triaxial tests were carried out. The cohesive strength and strength of the ISO sand and Liaoning soil used as the filling materials were measured. The test results are shown in Table 1.

**Table 1.** The values of cohesive strength and strength of ISO sand and Liaoning soil used as the filling materials.

| Filling Materials | Normal Stress (kPa) | Strength (kPa) | Cohesive Strength (kPa) |
|---|---|---|---|
| ISO sand | 100 | 125 | 0.20 |
| | 200 | 242 | |
| | 300 | 382 | |
| | 400 | 489 | |
| Liaoning soil | 100 | 109 | 37.60 |
| | 200 | 128 | |
| | 300 | 153 | |
| | 400 | 179 | |

The intact rock mass and cracked rock specimens containing one, two and three cracks were studied to investigate the effects of crack number on the dynamic damage of cracked rock, as shown in Figure 4a. Also, the cracked rock specimens with two cracks filled with sand, soil and the mixtures with the specified sand/soil ratios were studied to investigate the effects of filling material on the dynamic damage of cracked rock, as shown in Figure 4b. Each test group included three specimens to guarantee the repeatability of the experimental study.

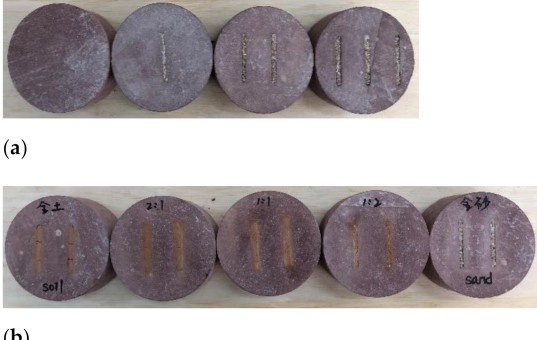

(a)

(b)

**Figure 4.** (**a**) Specimens with different numbers of cracks. (**b**) Specimens with different filling materials.

### 2.3. Typical Strain Gauge Records

The influence of the crack number and the filling material on the dynamic damage of cracked rock were investigated in this study. Incident, reflected and transmitted stress waves of the specimens with different numbers or different filling materials of cracks were obtained. A typical strain history recorded by the strain gauges is shown in Figures 5 and 6. Figure 5 shows the strain wave of the specimen with one crack and filled with sand. The waveforms of other specimens with different numbers of cracks were very similar. Figure 6 shows the strain wave of the specimen with two cracks filled with soil. The waveforms of other specimens with different filling materials had little difference. Since the impact velocity of the striker bar was controlled at almost the same value for each group, similar wave forms of incident waves were generated in the incident bar. The waveforms of the transmitted and reflected waves were very similar when the specimens were different. However, the amplitude of the strain wave was different, which was associated with the geometric characteristics of cracks.

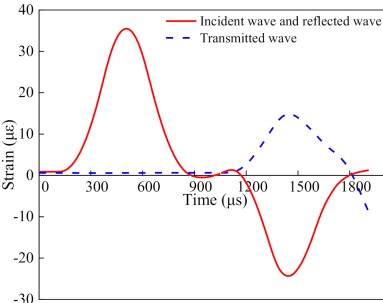

**Figure 5.** Typical waveforms of incident, reflected and transmitted pulses of the specimen with one crack filled with sand.

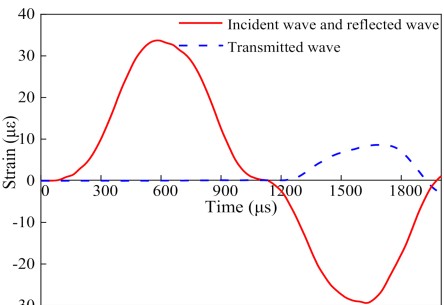

**Figure 6.** Typical waveforms of incident, reflected and transmitted pulses of the specimen with two cracks filled with soil.

Moreover, the dynamic force balance was critically assessed for the dynamic compressive tests in this study. According to the suggested method by the International Society for Rock Mechanics (ISRM), the pulse shaping technique placed at the free end of the incident bar (Figure 1) was applied to achieve the dynamic force balance. Based on the one-dimensional (1D) stress wave theory, the dynamic forces on the incident end ($P_1$) and the transmitted end ($P_2$) of the specimen are:

$$P_1 = AE(\varepsilon_{\mathrm{i}} + \varepsilon_{\mathrm{r}}), \qquad P_2 = AE\varepsilon_{\mathrm{t}} \tag{1}$$

Figure 7 shows the forces on both ends of the specimen in a typical dynamic compressive test. The forces ($P_1$ and $P_2$) on both ends of the specimen were approximately equal during the whole dynamic loading period (Figure 7). Consequently, the dynamic force balance on both loading ends of the specimen was achieved, i.e., $P_1 \approx P_2$.

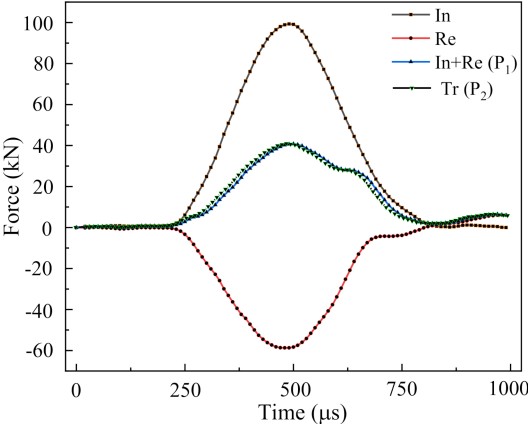

**Figure 7.** Dynamic force balance in typical dynamic compressive test. Note: In, force derived from the incident wave; Re, force derived from the reflected wave; Tr, force derived from the transmitted wave.

## 3. SHPB Test Results and Analysis

### 3.1. Energy Analysis

Based on one-dimensional stress wave theory, energy can be indirectly calculated according to the incident, reflected and transmitted stress wave signals obtained from the SHPB test. The calculation formula is shown as Equation (2) [38].

$$
\begin{aligned}
W_I &= \int_0^t A_i \sigma_i(t) c \varepsilon_i(t) dt = \frac{A_i c}{E} \int_0^t \sigma_i^2(t) dt \\
W_R &= \int_0^t A_r \sigma_r(t) c \varepsilon_r(t) dt = \frac{A_r c}{E} \int_0^t \sigma_r^2(t) dt \\
W_T &= \int_0^t A_t \sigma_t(t) c \varepsilon_t(t) dt = \frac{A_t c}{E} \int_0^t \sigma_t^2(t) dt
\end{aligned}
\tag{2}
$$

where $W_I, W_R, W_T$ are the incident wave energy, reflected wave energy and transmitted wave energy, respectively; $A_i = A_r = A_t = A$ represent the cross-sectional area of the bar; $E$ is the elastic modulus of the bar; $\sigma_i(t), \sigma_r(t), \sigma_t(t)$ are the incident, reflected and transmitted stress signals, respectively. The longitudinal wave velocity, $c$, can be calculated using the density, $\rho$, and the elastic modulus of the bar, as shown in Equation (3).

$$
c = \sqrt{E/\rho}
\tag{3}
$$

The stress wave can be deduced by strain signals, so Equation (2) can be expressed by Equation (4), as follows.

$$
\begin{aligned}
W_I &= AcE \int_0^t \varepsilon_i^2(t) dt \\
W_R &= AcE \int_0^t \varepsilon_r^2(t) dt \\
W_T &= AcE \int_0^t \varepsilon_t^2(t) dt
\end{aligned}
\tag{4}
$$

where $\varepsilon_i(t), \varepsilon_r(t), \varepsilon_t(t)$ are the incident, reflected and transmitted strain signals, respectively, and can be obtained from the SHPB test.

Ignoring the energy loss between the rock specimen and bars, the mathematical expression of total energy absorption is shown in Equation (5).

$$
W_d = W_I - W_R - W_T
\tag{5}
$$

Using the stress wave obtained from the records of strain gauges and employing Equations (4) and (5), the total absorption energy of the specimens with different numbers of cracks was obtained. The curves of absorption energy versus time of specimens with different numbers of cracks filled with sand are shown in Figure 8. It can be observed from Figure 8 that the tendency of the absorption energy was similar when the number of cracks was different. With the increasing number of cracks, the energy absorption increased. The experimental results suggested that cracked rock with more cracks usually absorbs more energy during its failure process, indicating that cracked rock with more cracks usually has a lower strength.

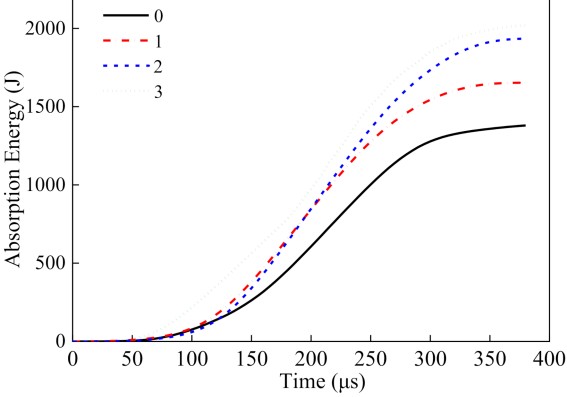

**Figure 8.** Curves of total absorption energy versus time of specimens with different numbers of cracks filled with sand.

Similarly, the total absorption energy of the specimens with different filling material was calculated. The curves of absorption energy versus time of specimens with two cracks filled with different materials are shown in Figure 9. Figure 9 indicates that the specimens with different filling materials have a similar tendency of the absorption energy. The filling material that had great strength and good cohesion could contribute more strength. As a result, the energy absorption decreased accordingly, indicating that the cracked rock filled with material having great strength and good cohesion holds the higher strength.

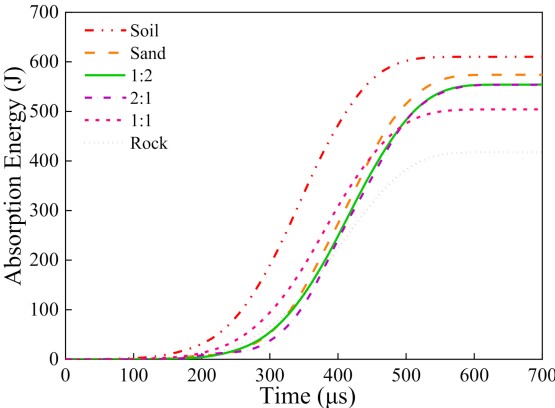

**Figure 9.** Curves of total absorption energy versus time of specimens with two cracks filled with different materials.

### 3.2. Dynamic Damage

According to the method proposed previously, the damage variable can be calculated using Equation (6) [39,40].

$$D = 1 - \frac{\sigma_j}{\sigma_r} \tag{6}$$

where $\sigma_j$ is the dynamic peak stress of cracked rock and $\sigma_r$ is the dynamic peak stress of intact rock mass.

The histories of stress of the specimen with different geometric characteristics can be obtained by analyzing the stress wave obtained from the records of strain gauges and can be derived as

$$\sigma(t) = \frac{A}{2A_0} E[\varepsilon_i(t) + \varepsilon_r(t) + \varepsilon_t(t)] \tag{7}$$

where $A_0$ is the initial cross-sectional area of the specimen. Assuming that the stress equilibrium condition or uniform deformation prevails during dynamic loading (i.e., $\varepsilon_i + \varepsilon_r = \varepsilon_t$) in Equation (7), Equation (7) can be rewritten as

$$\sigma(t) = \frac{A}{A_0} E\varepsilon_t(t) \tag{8}$$

The pulse shaping technique was used in this study to ensure the stress equilibrium. The force balance can be found in Figure 7.

Figure 10 shows the curves of stress versus time with different numbers of cracks. It can be seen that the stress form of specimens was very similar when the number of cracks was different. In addition, the peak stresses decreased with an increasing number of cracks. The largest stresses were 129.6, 125.8, 110.1 and 101.9 MPa when the number of cracks was zero, one, two and three, respectively.

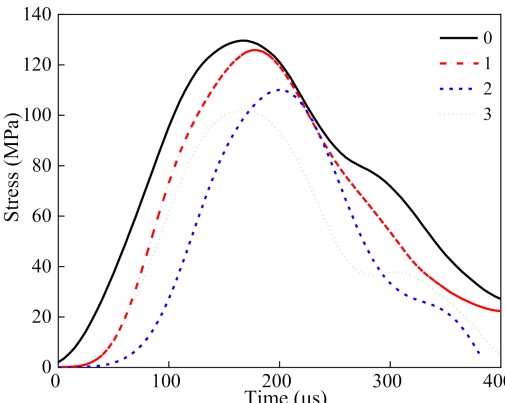

**Figure 10.** Curves of stress versus time of specimens with different numbers of cracks filled with sand.

According to Equation (6) and Figure 10, the damage variable of cracked rock with different numbers of cracks can be calculated (see Figure 11). The results showed that dynamic damage of cracked rock is obviously influenced by the number of cracks. The damage variable increased with an increasing number of cracks, which suggested that the cracked rock with more cracks was damaged more seriously under dynamic loading.

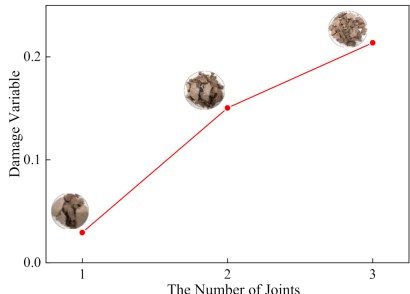

**Figure 11.** Curves of damage variable versus different number of cracks.

Further, it can be seen in Figure 12 that the stress form of specimens was very similar when the filling material was different. However, the peak stress was various. The extreme stress was 34.72, 57.33, 70.29, 58.63 and 46.98 MPa for the filling material being soil, 2:1 (the weight ratio of soil to sand), 1:1, 1:2 and sand, respectively. The corresponding stress was 77.52 MPa for the intact specimen. As shown in Figure 12, the peak stress of intact rock mass was bigger than that of the jointed rock mass. For the specimens with different filling materials, the order of the peak stress was 1:1 > 1:2 > 2:1 > sand > soil.

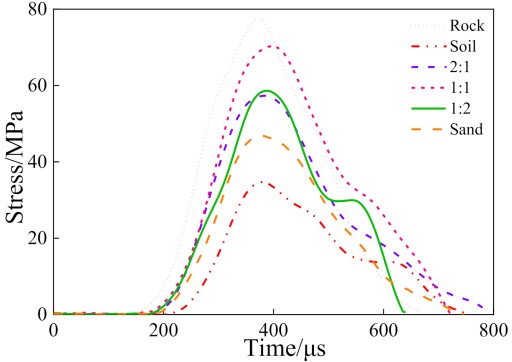

**Figure 12.** Curves of stress versus time of specimens with two cracks filled with different materials.

The damage variable of cracked rock with different filling materials can be calculated by Equation (6) and Figure 12, as is shown in Figure 13. The results suggested that the damage variable shown in Figure 13 was obviously influenced by the filling material. With the difference of the strength or cohesion of the filling material, the strength of cracked rock was various. The Liaoning soil and the ISO sand were chosen as initial materials in this study. Five types of filling materials were prepared: Liaoning soil, ISO sand, and the weight ratios of soil/sand = 2, 1, and 0.5. The cohesion of sand can be regarded as null. As a result, with the increasing ratio of sand, the cohesion of the filling material decreased. However, many studies have shown that the strength of sand is greater than that of soil. There is no doubt that the filling material with a higher ratio of sand had greater strength. As the strength of sand is greater than soil, the jointed rock filled with sand was harder than that filled with soil. At the same time, the cohesion of the filling material should be considered. The filling material that has great strength and good cohesion can contribute more strength. Considering the cohesion and strength of the filling materials (i.e., soil and sand) used in this study, the dynamic strengths of cracked rock can be ordered as follows: 1:1 (the weight ratio of soil/sand) > 1:2 > 2:1 > sand > soil.

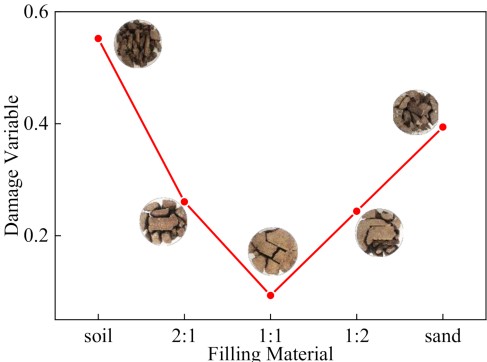

**Figure 13.** Curves of damage variable versus different filling materials.

## 4. Conclusions

In this study, dynamic compressive tests on sandstone specimens with different numbers of cracks and different filling materials were conducted using the split Hopkinson pressure bar (SHPB) apparatus. The energy absorption in this process was analyzed and the damage variable was obtained. The main conclusions are as follows:

1.  The SHPB test results demonstrate that the dynamic damage of cracked rock is obviously influenced by the geometric characteristics of cracks. With different spatial geometry, the energy absorption and the damage variable of the cracked rock is different during the impact process.
2.  The energy absorption and damage variable increased with the increasing number of cracks, which suggests that cracked rock with more joints usually holds a lower strength.
3.  The different strengths or cohesion of the filling material resulted in different strengths of cracked rock. The filling material that has great strength and high cohesion can contribute more strength. As a result, the corresponding energy absorption and dynamic variable decrease accordingly. The cracked rock filled with material with great strength and good cohesion always has a higher strength.

**Author Contributions:** F.W., writing—original draft preparation. S.W. and Z.X., writing—review and editing. Z.X. and F.W., testing investigation and theoretical calculation. All authors have read and agreed to the published version of the manuscript.

**Funding:** This research was supported by National Natural Science Foundation of China (Grant Nos. 51474050 and U1602232); Liaoning Science and Technology Project (2019JH2/10100035) and the China Scholarship Council (CSC) (201806080103).

**Informed Consent Statement:** Informed consent was obtained from all subjects involved in the study.

**Data Availability Statement:** The data presented in this study are available on request from the corresponding author.

**Conflicts of Interest:** The authors declare that they have no conflict of interest.

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
