# Peer review of "Influence of Crack Geometry on Dynamic Damage of Cracked Rock: Crack Number and Filling Material"

_applsci, doi:10.3390/app11010250_

Round 1

Reviewer 1 Report

Review Report (MS# ID: applsci-1020552)

Dynamic damage of cracked rock with different geometric characteristics

by

Feili WANG, Shuhong WANand Zhanguo XIU

The manuscript evaluates the energy dissipation (or total energy absorption) and the damage variable for rock or sandstone with different number of cracks and different filling materials using the conventional split Hopkinson pressure bar(SHPB). The test results indicate that both the energy dissipation and the damage variable increase with increasing number of cracks, depending on the filling materials, and that the peak stress decreases with increasing number of cracks.

Overall, the content of the manuscript is sound and interesting. A survey of relevant references on the subject is well made. However, the manuscript lacks in many important descriptions. The title does not exactly reflect the content of the manuscript. The data reduction by the Hopkinson bar analysis is not described in details. The test data included are of minor value, and their contribution to the readers in this field is small. Since the dynamic compression tests or the SHPB tests are successfully performed, the effect of strain rate on the dissipation energy and the damage variable should be elucidated. Unfortunately, the reviewer believes that the manuscript should not be accepted for publication as its present form. The authors are asked to address the following comments to improve the manuscript:

Mandatory comments:

  1. Line 63 (Section 2.1): The dimensions and material of the striker bar should be given. The specifications for the strain gauges mounted on the incident and transmitter bars should be provided, e.g., gauge length, gauge factor and so on. The measurement system including the oscilloscope and the sampling rate should also be explained.
  2. Line 68 : Is the P-wave velocity used in the same meaning of the longitudinal wave velocity in Eq.(2) or not ?
  3. Line 88 (Section 2.1): The static compressive stress-stain curves of the specimen with and without out cracks (one crack is alright) should be given for comparison with the corresponding dynamic ones with and without crack.
  4. Line 120 and Line 122 (or Figure captions of Figs. 4 and 5): “Stress” wave in the two Figure captions is erroneously used, because the title of the vertical axis in both Figs. 4 and 5 indicates the strain. Therefore, “Strain” wave should be used.
  5. Line 147(Fig.6), Line 149 (Fig.7), Line 168(Fig.8), Line 184 (Fig.10): The total energy absorption, the stress history, and the damage variable should be expressed using a parameter of the average strain rate up to the final time. In Figs. 6 and 7, how did you determine the final time in the time-integration by Eq.(4)?
  6. Line 159: In Eq.(6), sr denotes the peak stress of the intact rock mass. Does this indicate the static or dynamic stress ?
  7. Line 168: In Fig. 8 (also in Fig.10) are shown the stress histories for specimens with different number cracks. The expression for determining the stress histories in the specimen should be provided. If the expression under the assumption of dynamic force equilibrium within the specimen is used, the validity of the expression should be verified.

Suggestions

  1. Several sentences do not make any sense. Misuse of the terminology is found in some parts of the text. English language editing should be done.
  2. In each graph, the first letter of the titles of both the vertical and horizontal axes should be capitalized.

Reviewer 2 Report

The paper used the SHPB apparatus to investigate cracked rocks with different filling materials. The authors need to improve the paper substantially before it can be considered for publication. Below are my specific comments:

1) the paper needs to be checked for language and spelling. The authors have used "very" instead of "vary" many times in the paper.

2) there are multiple occasions where authors introduce new concepts and used complex statements that are difficult to follow. Lines 17-19, 40-42 are examples of such complex statements that do not make sense. Authors need to rewrite these.

3)Details of specimen preparation are missing. How was crack introduced in the sandstone? How was filler material added to crack? was any filler compaction procedure followed? These details are essential and need to be provided.

4)Fig 4 and 5 should be modified to have the same size and same axis spans.

5) Multiple figures have gray colored curves that are difficult to view. Replace these with a more visible color.

6)Authors need to specify how many cracks were there in the specimens used for fig7 and fig 11. 

7) the difference between 2:1 and 1:2 ratio is not clear. Add details of filler material ratios and terminology used to address these ratios in the materials section.

8)Authors need to add more discussion about clay and sand on strength and cohesivity.

Round 2

Reviewer 1 Report

Review Report (MS ID: applsci-1020552R1)

Influence of crack geometry on dynamic damage of cracked rock

: crack number and filling material

by

Feili WANG, Shuhong WANand Zhanguo XIU

In response to the reviewer’s comments and suggestions, the manuscript has been greatly revised and improved. The title and Abstract were appropriately revised, and a new Figure 2 was added to explain the measurement system. The experimental procedures including the specifications for the strain gages and the dimensions of the striker were described in more detail and the formula Eq.(8) for evaluating the stress applied to the rock specimens was further provided. The legends in Fig. 5 and Fig.6 for the output from the incident gage were also revised. It is worth noting that insert pictures of fractured specimens were added in Fig. 10 and Fig.12 to indicate the fracture modes in the specimens after dynamic testing. However, the reviewer has several reservations on the text. The authors are asked again to address the following comments to improve the manuscript:

Mandatory comments:

  1. Line 91- Line 92: It is advisable to explain if a lubricant is used or not before the specimen is sandwiched between the incident and transmitter bars. This is because Line 158 describes the neglect of the (frictional) energy loss between the rock specimen and the bars.
  2. Line 128-Line 129: the stress wave should be replaced with the strain wave
  3. Line 133, similar wave forms of incident waves : The duration of the incident strain waves shown in Fig.5 and Fig.6 is different from each other. The duration depends only on the length of the striker, not depending on the specimen material or the impact velocity of the striker. The same pulse shaper is supposed to be used in this study. What is the reason for this ?
  4. Line 149: Ai =Ar =At should be expressed as Ai =Ar =At =A, because A is used later in Eq.(3) and Eq.(4). Accordingly, Line 188 or, “where A is cross-sectional area of the bar” can be deleted.
  5. Line 153: Eq.(3) is not practically used in SHPB analysis and then can be omitted, because Eq.(4) can be derived directly from Eq.(1) without Eq.(3) using Hooke’s law or s = E e.
  6. Line 190 : The validity of the stress equilibrium in the specimen (ei + er = et) should be verified using the SHPB test data, because Eq.(8) derived from the stress equilibrium condition is applied to determine the stress histories for the rock specimens in Fig. 9 and Fig.11.

Suggestions

  1. Misuse of the terminology is still found in some parts of the text.

For example:

 P-wave velocity (Line 69) and longitudinal wave velocity (Line 151) are used in common.

“transmitted” bar (in Figure 1 on Line 71 ) and “transmission” bar (Line 78) are used in common.

Line 150 and Line156: incident, reflection and transmission stress or strain signals should be written as incident, reflected and transmitted stress or strain signals

  1. Several words in the text do not make any sense. English language editing is needed.

For example:

 Line 105: rectangular fracture ?  rectangular crack sizes

Line 166: Which

Line 202 : the Resulted

and so on.

Reviewer 2 Report

The authors have made significant improvements to the paper. However, some more additional information still needs to be added.

1)The authors need to add a description of the procedure to introduce the crack in a specimen. How was crack introduced? What machine was used?

2) Authors need to provide numerical values (range) of cohesive strength and strength of soil and sand used based on literature studies.
